# Prevalence and factors associated with acute respiratory infection among under-five children in selected tertiary hospitals of Kathmandu Valley

Pratima Ghimire[1][*], Rashmi Gachhadar[2], Nebina Piya[1], Kunja Shrestha[1], Kalpana Shrestha[1]

1 Department of Nursing, Nepal Medical College Pvt. Ltd.,/Kathmandu University, Kathmandu, Bagmati Province, Nepal, 2 Maharajgunj Nursing Campus, Institute of Medicine/Tribhuvan University, Kathmandu, Bagmati Province, Nepal

☯ These authors contributed equally to this work.

* ppratima071@gmail.com

**Data Availability Statement:** All relevant data are within the paper and its supporting information files.

## Abstract

### Background

Acute respiratory infection (ARI) is responsible for about 30–50 percent of visits to health facilities and for about 20–30 percent of admissions to hospitals in Nepal for children under 5 years old. Incidence of ARI in children among under-five years of age is 344 per 1000 in Nepal. Hence, the study aims to find out the prevalence and factors associated with acute respiratory infection among under-five children.

### Methods

A cross-sectional study was conducted at Nepal Medical College and Teaching Hospital and International Friendship Children's Hospital (IFCH) in Kathmandu among children of age 2–59 months attending Pediatric OPD. A total of 286 children were selected using the non-probability (convenient) sampling technique. Data were collected using pre-tested semi-structured tool through interview schedule, and descriptive and inferential statistical analyses were used.

### Results

Out of 286 children, more than half of children (60.8%) had Acute Respiratory Infection (ARI). Nearly one-fifth of the children had severe or very severe pneumonia. Acute respiratory infection was significantly associated with religion followed by the family (p = 0.009, OR = 4.59 CI = 1.47–14.36), presence of the child in the kitchen while cooking (p = 0.001, OR = 2.03 CI = 1.17–3.51), and presence of respiratory tract infection in family (p = <0.001 OR = 2.83 CI = 1.59–5.05).

**Funding:** This research was supported by Nepal Medical College-Institutional Review Committee (NMC-IRC): Ref.009-075/076 https://nmcth.edu/category/research/research_committee The funders had no role in study design, data collection and analysis, decision to publish, or preparation of the manuscript.

**Competing interests:** The authors have declared that no competing interests exist.

## Conclusion

The study concluded that male children are more susceptible to acute respiratory infection than female children. Parents and family members should be aware of the prevention of acute respiratory infection by addressing and minimizing the factors contributing to ARI.

## Introduction

Acute respiratory infections (ARIs) are infections of the airways from nostrils to the alveoli. ARIs can be categorized as upper respiratory tract infections (URIs) or lower respiratory tract infections (LRIs) [1]. Acute respiratory infection poses a major challenge to the health system, especially in developing countries and is the leading cause of mortality and morbidity among children under five years [2, 3].

According to the World Health Organization (WHO), respiratory infections account for 6% of the total global disease burden. Around 6.6 million, under-five aged children years of age die each year worldwide; 95 percent of them belong to low-income countries and one third of the total deaths is due to ARI [4]. It is estimated that Bangladesh, India, Indonesia and Nepal together account for 40% of the global ARI mortality. ARI is responsible for about 30–50% of visits to health facilities and for about 20–40% of admissions to hospitals for under-five children [3].

The Nepal Demographic and Health Survey (NDHS) 2016 revealed that 31% of the causes of neonatal mortality in Nepal were respiratory and cardiovascular disorders [5]. In fiscal year 2074/75, a total of 17, 50, 668 ARI cases among under-five were registered in Nepal, out of which 10.5% were categorized as pneumonia cases and 0.29% were severe pneumonia cases. ARI case fatality rate per thousand for under-five children at the health facility is 0.05 in Nepal [6].

Acute respiratory infection is linked to various modifiable risk factors including demographic, environmental, socio-economic, and nutritional factors [7]. Many studies have shown that comorbid illnesses especially HIV, malnutrition, prematurity or measles, family history of ARI, low socioeconomic status, inappropriate weaning time, pallor, severe malnutrition and cooking fuel other than liquefied petroleum gas, indoor air pollution, maternal illiteracy, parental smoking behavior male gender, rural residency and overcrowding associated with ARI [2, 7–9]. If the associated modifiable risk factors could be modified and/or avoided through the implementation of various intervention strategies then, the disease burden in the community could be reduced [7].

Despite the burden of acute respiratory infection on morbidity and mortality in children under-five children in the world, there is limited data to evaluate the problem in Nepal. The availability of data on the prevalence and risk factors of ARIs is vital because achieving Sustainable Development Goal on improving health and wellbeing will depend on the existing efforts to prevent and control ARIs in all WHO regions. Many socio-cultural, demographic, and environmental risk factors predispose children less than 5 years to acquire Respiratory Tract Infections (RTIs) [4].

Thus, this study aims to determine the prevalence and factors associated with acute respiratory infection among under-five children in selected tertiary hospitals of Kathmandu Valley.

## Materials and methods

### Aim, design, and setting

A cross sectional study was carried to assess the prevalence of Acute Respiratory Infection (ARI) and to identify the factors associated with ARI at two tertiary level hospitals of

Kathmandu Valley; Nepal Medical College Teaching Hospital and International Friendship Children's Hospital.

## Study participants

Children attending the Pediatric Out Patient Department (OPD) of Nepal Medical College Teaching Hospital and International Friendship Children's Hospital were selected as study participants. The inclusion criteria included: children aged ≥2–59 months visiting the hospital OPD for either respiratory or any other problems. The exclusion criteria were; children with clinically diagnosed bronchial asthma (history of repeated episodes of wheeze with rapid response to bronchodilator and children with any co-morbidity or any other physical and/or intellectual disabilities.

## Sampling technique and sample size

A convenient sampling technique was adopted for this study. A similar study conducted in Gorkha Municipality of Nepal reported the overall prevalence of ARI to be 21.5% [10]. So, using this prevalence with an allowable error of 5% at a confidence level of 95%, the sample size was estimated to be 286 participants after adding a non-response rate of 10%.

## Instruments

Pretested semi-structured questionnaire was used which was developed after extensive literature review and consulting with subject expert.

The tool was translated into Nepali language and back translated into English; then Nepali version of tool was pretested among 29 participants (10.14% of the total participants) for validation before the final administration. The reliability of the tool was calculated using Cronbach's alpha, which was 0.85.

The research instrument consists of the following parts:

Part I: Socio-demographic characteristics of the child's family

It includes information as age of the attendee, gender of the attendee, caste/ethnicity and religion of children's family; information on the type of family, family size, and socio-economic status (as per modified Kuppuswamy scale) [11].

Part II: Environmental Characteristics

It includes information as the type of house, number of family members per room, cross-ventilation, place of cooking, presence of a child in kitchen while cooking and history of smoking in a family within the living areas.

Part III: Child related Information

It includes information on child as age and sex of the child, birth status, birth weight, anthropometric measurement (height and weight), exclusive breast-feeding, the month of weaning, and immunization status

Part IV: Prevalence of ARI

The prevalence of ARI was assessed using revised WHO classification of childhood pneumonia at health facilities [12] which is classified as no pneumonia, pneumonia and severe or very severe pneumonia where, children with cough and cold were classified as no pneumonia, children with fast breathing and/or chest wall indrawing were classified as pneumonia, and children with cough and difficult breathing and/or chest wall indrawing and presence of general danger signs, such as history of convulsions, inability to feed, incessant vomiting and lethargy or unconsciousness were classified as severe or very severe pneumonia.

**Variables under the study.** *Dependent variable*: *Acute Respiratory Infection (ARI)*. Independent variables: age and sex of the child, birth status, birth weight, caste/ethnicity, religion,

type of family, family size and socio-economic status, type of house, number of family member per room, cross-ventilation, place of cooking, presence of child in kitchen while cooking, history of smoking in family, exclusive breast-feeding, month of weaning, immunization status and nutritional status

### Procedure of data collection

The data was collected from March, 2019 to June, 2019 in the OPD of Nepal Medical College Teaching Hospital and International Friendship Children's Hospital through interview schedule after obtaining formal permission from the respective hospitals. The attendees of the children were asked about the general information about the child and child's family. The attendees were either mother or father of the child. The procedure and purpose of the study were explained to the attendees and were recruited based on inclusion and exclusion criteria. Those willing to participate were interviewed in the waiting area of OPD. Informed consent was read to the attendees and once signed, a face to face interview was conducted using the pretested semi-structured questionnaires by the researchers themselves. After obtaining general information, anthropometric measurement i.e. height and weight of the child was taken and finally the child was assessed for the presence of ARI as per WHO criteria. Children were recruited until the sample size was met.

### Data management and statistical analysis

The collected data was entered into Microsoft Excel and coded with alphanumerical codes which were finally converted into Statistical Package for Social Sciences (SPSS) for statistical analysis. Continuous variables were categorized using their mean value. Bi-variate analysis (Simple Logistic regression) was used to show relationship between the categorical variables. Variables which had $p<0.2$ were subsequently put on stepwise multiple logistic regression model to determine the significant independent risk factor of ARI.

### Ethical consideration

The study was approved by the Institutional Review Committee of Nepal Medical College. Permission for data collection was obtained from the Hospital Director and Head of the Department of Pediatric OPD of the respective hospitals. Written informed consent was taken from the parent (Mother or Father) of children attending pediatric OPD prior to data collection and all the information were kept confidential.

## Results

A total of 286 children were recruited in the study. Interview was carried with the parents, of which most of them were mothers or fathers of the child, while some of them were caregivers. More than half of the children (64%) were aged 24 months and above and relatively similar percentage of the children (62.2%) were male, 11.5% of the child were born preterm while 7.3% them were child with low birth weight. Regarding nutritional status, most of the children (97.9%) were of normal nutritional status and 60.5% of the children were exclusively breastfed. More than half of the children (59.4%) were weaned after six months of age while, most of them (96.2%) had completed immunization as per their age. Majority of the children's family (73.4%) followed Hindu religion and more than 3/4th of the children belong to middle class family. Majority (86%) of the children lived in pucca house, 24.1% lived in over-crowded house and 23.1% had poor ventilation in their house. More than one fourth of the family had their kitchen within living room or same room, 54.2% of the children were carried on back by

**Table 1. Prevalence of ARI.** n = 286.

| Characteristics | Frequency(n) | Percentage (%) |
|---|---|---|
| **No pneumonia** | 87 | 30.4 |
| **Pneumonia** | 46 | 16.1 |
| **Severe or very severe pneumonia** | 41 | 14.3 |
| **Overall prevalence of ARI** | 174 | 60.8 |

their mother during cooking and 32.2% of the children's family member were involved in smoking.

More than half of the children (60.8%) of the children were found to be have acute respiratory infection including 14.3% being diagnosed as severe or very severe pneumonia. [Table 1]

Bivariate analysis of the factors shows that the variables like religion followed by the family, presence of child in the kitchen while cooking and presence of respiratory tract infection were significantly associated with ARI. [Table 2]

Table 3 shows the multivariable model where the variables significantly associated were religion, presence of the child in kitchen while cooking and presence of RTI in family. From the odds ratio evaluation, the odds of having ARI among Buddhist and others were higher (4.59 times and 6.12 times respectively) than the Hindu. The probability of ARI was 2.83 times higher among the children whose family members had history of RTI. Similarly, the probability of ARI was double in the children if they were present inside kitchen while cooking. [Table 3]

## Discussion

### Prevalence of Acute Respiratory Infection (ARI)

The overall prevalence of ARI was 60.8%. The finding of this study is higher than the national data i.e. 2% of under-five children are affected with ARI in Nepal [13]. The prevalence is also higher than a study done in Gorkha Municipality, Nepal which showed 21.5% prevalence of ARI among under-five children [10]. While, one of the study conducted in Pokhara, Nepal showed 63% of the under-five children had ARI which is higher than the prevalence of current study [14]. Various studies conducted in India found lower prevalence (22%-52%) of ARI [7–8, 15–19]. On the contrary, study done in Shivrajpur, India showed higher prevalence i.e. 79.02% than the current study [3]. The findings are also higher than the studies done in Cameron (54.7%), Ethiopia (27.3%-33.5%) and Swat (29%) [4, 20–22].

Among the children who had symptoms of ARI, half of them had no pneumonia, 26.44% had pneumonia and 23.56% had severe or very severe pneumonia, while one of the study done in India found pneumonia among 23.3%, severe pneumonia among 47.7% and very severe pneumonia among 29% of the children and the study carried out in Cameron found 59% prevalence of no pneumonia, 25% pneumonia and 16% severe pneumonia [4, 7].

### Factors associated with ARI among under-five children

The multivariate analysis showed, the variables significantly associated were again religion, presence of the child in kitchen while cooking and presence of RTI in family. From the odds ratio evaluation, the odds of having ARI among children following Buddhism and others were higher (p = 0.009, OR = 4.59 CI = 1.47–14.36) and (p = 0.004, OR = 6.12 CI = 1.77–21.15) respectively than the children following Hinduism. Therefore, children whose family was following buddhism religion were 4.5 times and other than hinduism and buddhism were 6 times likely to have ARI. The reason behind this might be the genetic inheritance. Children

**Table 2. Bivariate analysis of the factors associated with ARI.** n = 286.

| Variable | Total | ARI present | ARI absent | p-value | Unadjusted Odds Ratio with 95% C. I. |
|---|---|---|---|---|---|
| **Age of the child** | | | | | |
| ≤24 months | 183 | 114 | 69 | 0.502 | 0.85 (0.52–1.38) |
| >24 months | 103 | 60 | 43 | | |
| **Sex of the child** | | | | | |
| Male | 178 | 111 | 67 | 0.499 | 0.85 (0.52–1.38) |
| Female | 108 | 63 | 45 | | |
| **Birth weight of the child** | | | | | |
| Normal or above | 265 | 163 | 102 | 0.412 | 0.69 (0.28–1.68) |
| Low birth weight | 21 | 11 | 10 | | |
| **Birth status of the child** | | | | | |
| Term | 253 | 155 | 98 | 0.683 | 1.17 (0.56–2.43) |
| Pre-term | 33 | 19 | 14 | | |
| **Immunization status of the child** | | | | | |
| Complete as per age | 275 | 168 | 107 | 0.664 | 0.76 (0.23–2.57) |
| Incomplete as per age | 11 | 6 | 5 | | |
| **Nutritional status (W/A)** | | | | | |
| Normal | 280 | 172 | 108 | 0.185 | 0.31 (0.06–1.74) |
| Under-nutrition | 6 | 2 | 4 | | |
| **Exclusive Breastfeeding** | | | | | |
| Exclusively breastfed | 173 | 101 | 72 | 0.293 | 1.30 (0.79–2.12) |
| Non-exclusively breastfed | 113 | 73 | 40 | | |
| **Month of weaning** | | | | | |
| After six months | 170 | 98 | 72 | 0.181 | 1.39 (0.86–2.28) |
| Before six months | 116 | 76 | 40 | | |
| **Number of siblings** | | | | | |
| ≤1 | 183 | 117 | 66 | 0.154 | 0.69 (0.43–1.14) |
| >1 | 103 | 57 | 46 | | |
| **Religion followed by family** | | | | | |
| Hindu | 210 | 127 | 83 | 0.047* | |
| Buddhist | 59 | 41 | 18 | 0.050 | 2.81 (0.99–7.88) |
| Others | 17 | 6 | 11 | 0.014* | 4.18 (1.34–13.04) |
| **Type of family** | | | | | |
| Nuclear | 142 | 91 | 51 | 0.265 | 0.76 (0.47–1.23) |
| Joint or extended | 144 | 83 | 61 | | |
| **Educational status of the mother** | | | | | |
| Iliiterate | 13 | 8 | 5 | 0.802 | 0.96 (0.69–1.33) |
| Literate | 273 | 166 | 107 | | |
| **Educational status of the father** | | | | | |
| Illiterate | 6 | 4 | 2 | 0.968 | 1.01 (0.71–1.43) |
| Literate | 280 | 170 | 110 | | |
| **Occupational status of the mother** | | | | | |
| Unemployed | 223 | 135 | 88 | 0.884 | 1.06 (0.59–1.88) |
| Employed | 63 | 39 | 24 | | |
| **Occupational status of the father** | | | | | |
| Unemployed | 10 | 5 | 5 | 0.254 | 1.30 (0.83–2.05) |
| Employed | 276 | 169 | 107 | | |
| **Socio-economic status of the family** | | | | | |

(*Continued*)

**Table 2.** (Continued)

| Variable | Total | ARI present | ARI absent | p-value | Unadjusted Odds Ratio with 95% C. I. |
|---|---|---|---|---|---|
| Upper class | 20 | 15 | 5 | 0.260 | 0.75 (0.45–1.24) |
| Middle class | 219 | 132 | 87 | | |
| Lower class | 47 | 27 | 20 | | |
| **Type of house** | | | | | |
| Kutcha and semi-pucca | 40 | 24 | 16 | 0.907 | 1.04 (0.53–2.06) |
| Pucca | 246 | 150 | 96 | | |
| **Place of cooking (Kitchen)** | | | | | |
| Within bedroom or sitting room | 74 | 49 | 25 | 0.272 | 0.73 (0.42–1.28) |
| Separated | 212 | 125 | 87 | | |
| **Type of cooking fuel used** | | | | | |
| LPG | 226 | 164 | 102 | 0.307 | 0.62 (0.25–1.55) |
| Firewood and others | 20 | 10 | 10 | | |
| **Presence of the child in kitchen while cooking** | | | | | |
| Yes | 155 | 108 | 47 | 0.001* | 0.44 (0.27–0.72) |
| No | 131 | 66 | 65 | | |
| **Parental smoking** | | | | | |
| Yes | 92 | 57 | 35 | 0.790 | 0.93 (0.56–1.55) |
| No | 194 | 77 | 117 | | |
| **Adequacy of cross-ventilation** | | | | | |
| Adequate | 220 | 129 | 91 | 0.165 | 1.51 (0.84–2.71) |
| Inadequate | 66 | 45 | 21 | | |
| **Over-crowding** | | | | | |
| Yes | 69 | 45 | 24 | 0.393 | 0.78 (0.44–1.38) |
| No | 217 | 129 | 88 | | |
| **Presence of RTI in family** | | | | | |
| Yes | 100 | 77 | 23 | <0.001* | 3.07 (1.78–5.31) |
| No | 186 | 97 | 89 | | |
| **Presence of RTI in siblings** | | | | | |
| Yes | 60 | 39 | 21 | 0.458 | 0.79 (0.44–1.45) |
| No | 226 | 135 | 91 | | |

(RTI = Respiratory Tract Infection; W/A = Weight per Age of the child)

*p-value significant at 95%

who were exposed to the family members (other than sibling) with RTI had 2.8 times probability of developing the symptoms of ARI. Similar finding have been identified by the studies done in Nepal (children are 7 times more likely to develop ARI when exposed to family member with symptoms of RTI), India (children are 5.32 times likely to develop ARI when exposed to family member with symptoms of ARI), Cameron (children are 3.37 times likely to develop ARI when exposed to family member with symptoms of ARI) and Zambia (children are 2.3 times likely to develop ARI when exposed to family member with symptoms of ARI) [2, 5, 9, 20]. A hospital based case-control study carried in India also found children whose mother had presence of Upper Respiratory Tract Infection (URTI) had 6.5 times were more likely to have ARI and presence of Lower Respiratory Tract Infection (LRTI) among family members were 5.15 times more likely to have ARI [23]. Children who were present in the kitchen during the time of cooking tend to develop symptoms of ARI double then those who were not present.

**Table 3. Multivariate analysis of the factors associated with ARI.** n = 286.

| Variables | β Coefficient | p-value | Odds Ratio with 95% C. I. |
|---|---|---|---|
| **Religion** | | | |
| Hindu | | 0.016 | Ref |
| Buddhist | 1.526 | 0.009 | 4.59 (1.47–14.36) |
| Others | 1.812 | 0.004 | 6.12 (1.77–21.15) |
| **Number of siblings** | | | |
| ≤1 | | | Ref |
| >1 | 0.429 | 0.119 | 1.54 (0.89–2.63) |
| **Presence of RTI in family** | | | |
| No | | | Ref |
| Yes | 1.042 | <0.001 | 2.83 (1.59–5.05) |
| **Adequacy of cross-ventilation** | | | |
| Inadeqaute | | | Ref |
| Adequate | -0.146 | 0.665 | 0.86 (0.45–1.68) |
| **Presence of the child in kitchen while cooking** | | | |
| No | | | Ref |
| Yes | 0.71 | 0.012 | 2.03 (1.17–3.51) |
| **Month at weaning** | | | |
| Six months or before | | | Ref |
| After six months | -0.344 | 0.204 | 0.71 (0.42–1.21) |
| **Nutritional Status** | | | |
| Normal | | | Ref |
| Underweight or overweight | 0.905 | 0.371 | 2.47 (0.34–17.95) |

A study carried in Bamenda Regional Hospital, Cameron also found children were 1.85 times more prone to ARI if they were exposed to wood smoke [4].

The present study shows no significant association of age, sex, birth weight, immunization status, nutritional status, exclusive breastfeeding, number of siblings, presence of ARI in sibling, type of family, parental education and occupation, socio-economic status, overcrowding, cross-ventilation and type of house. In contrast to this finding, various studies have identified these factors to be significantly associated with ARI among under-five children [2–4, 7–10, 14–22, 24–27].

## Conclusions

Acute Respiratory Infection (ARI) is a major problem among under-five children. The prevalence of ARI among the male child is more than that of female child. The prevalence was higher among the children of age less than 24 months. Children when get exposed to smoking while cooking have greater chance of acquiring ARI. Similarly, when the family members have respiratory tract infections, it could double the risk of ARI among the children. Larger studies with case-control design in the community settings are recommended to find out the major factors of ARI. Children should not be exposed to fuel smoking unnecessarily and parents need to be more cautious when any of the family members have respiratory tract infection.

## Supporting information

**S1 Table. Socio-demographic information of the child.** n = 286.
(TIF)

**S2 Table. Distribution of child related environmental characteristics.** n = 286.
(TIF)

**S3 Table. Health information of the child.** n = 286.
(TIF)

## Acknowledgments

Researcher would like to thank the parents of children who shared their valuable experiences, spent precious time and for their participation and the faculties of Pediatric Department of Nepal Medical College and Teaching Hospital.

## Author Contributions

**Conceptualization:** Pratima Ghimire.

**Formal analysis:** Pratima Ghimire, Rashmi Gachhadar, Nebina Piya, Kunja Shrestha, Kalpana Shrestha.

**Investigation:** Pratima Ghimire, Rashmi Gachhadar, Nebina Piya, Kunja Shrestha.

**Methodology:** Pratima Ghimire.

**Resources:** Pratima Ghimire.

**Software:** Rashmi Gachhadar.

**Supervision:** Kalpana Shrestha.

**Validation:** Pratima Ghimire, Rashmi Gachhadar.

**Writing – original draft:** Pratima Ghimire, Rashmi Gachhadar, Nebina Piya.

**Writing – review & editing:** Pratima Ghimire, Rashmi Gachhadar, Nebina Piya.

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
