## [Decision Letter · Decision Letter 0]

26 Nov 2021

PONE-D-21-19925Prevalence and Factors Associated with Acute Respiratory Infection among Under-five Children in Selected Tertiary Hospitals of Kathmandu ValleyPLOS ONE

Dear Dr. Pratima Ghimire,

Thank you for submitting your manuscript to PLOS ONE. After careful consideration, we feel that it has merit but does not fully meet PLOS ONE’s publication criteria as it currently stands. Therefore, we invite you to submit a revised version of the manuscript that addresses the points raised during the review process.

Thank you for your interesting article, but it needs Major Revision following review by expert's reviewers. Please could you address the concerns raised from the reviewers and re-submit the revised version. 

We look forward to receiving your revised manuscript.

Kind regards,

Prof Sajid Bashir Soofi

Academic Editor

PLOS ONE

Journal Requirements:

Additional Editor Comments (if provided):

Thank you for submitting your manuscript to PLOS ONE. After careful consideration, we feel that it has merit but does not fully meet PLOS ONE’s publication criteria as it currently stands and need Major Revision. Therefore, we invite you to submit a revised version of the manuscript that addresses the points raised during the review process by the reviewers.

Reviewers' comments:

Reviewer's Responses to Questions

**Comments to the Author**

1. Is the manuscript technically sound, and do the data support the conclusions?

Reviewer #1: No

Reviewer #2: Yes

2. Has the statistical analysis been performed appropriately and rigorously? 

Reviewer #1: Yes

Reviewer #2: Yes

3. Have the authors made all data underlying the findings in their manuscript fully available?

Reviewer #1: No

Reviewer #2: No

4. Is the manuscript presented in an intelligible fashion and written in standard English?

Reviewer #1: No

Reviewer #2: Yes

5. Review Comments to the Author

Reviewer #1: The manuscript is not presented in an intelligible fashion and is written in standard English. Authors need to read this paper carefully and make all the necessary changes, as there are several errors while reporting this paper.

Reviewer #2: Abstract

A few clarifications needed. Please reword:

Acute respiratory infection (ARI) is responsible for 30-50 percent of visits to health facilty and 20-30 percent of admissions to hospital in Nepal. – is this for both adults and children or for children under 5 years old?

What does it mean, ‘incidence of ARI/1000 children < 5 years of age is 344 in Nepal’? Does this mean 344/1000? Please reword: The incidence of ARI in children < 5 years old is 344/1000.

The sentences are not clear. Please reword to make it clear.

Non-probability is not equivalent to purposive. Purposive is to target those with ARI. In this study, I believe it is convenient sampling – please stick to convenient sampling.

When you say out of the 286 children, half had ARI. What did the other children have?

Also, do not include conclusion that male children are not susceptible to ARI, because the results were insignificant.

Introduction

Line 60: ARI is responsible for about 30-50% of visits…. Question: is this in children under 5 years old? Please specify.

Line 64: The deaths due to ARI at health facility is 127. What does this mean? Any health facility or hospitals or primary care? Can remove this information as it is not clear.

Methodology

Line 91: Please specify the time and duration of the data collection period e.g. from month, year until month, year.

Line 106: Please add the Cronbach alpha for the pretesting that you have done, to show whether the questionnaire was reliable.

From paragraph 66-73 in the introduction, please add in the methodology the factors that was studied in your study.

Chi-squared test was done to show relationship between the categorical variables. My personal opinion is it is better to do simple logistic regression, and then move on to multiple logistic regression for the significant factors. Any reason why you use Chi-squared test rather than simple logistic?

Results – okay

Discussion

You are right to discuss in line 204 regarding the prevalence of ARI.

218: In the discussion, you do not need to mention twice but to just discuss about the three factors that were associated with increase risk of ARI in children, which are religion, presence of child in the kitchen while cooking and presence of respiratory tract infection in family member.

6. PLOS authors have the option to publish the peer review history of their article (what does this mean?). If published, this will include your full peer review and any attached files.

Reviewer #1: No

Reviewer #2: **Yes: **Associate Professor Dr. Farnaza Ariffin

---

## [Author Response · Author response to Decision Letter 0]

6 Jan 2022

Response to Academic editor 

Author Response: The manuscript has been revised and corrected as per the requirements of PLOS ONE.

2. In your Data Availability statement, you have not specified where the minimal data set underlying the results described in your manuscript can be found.

Author Response: All the data are with in the manuscript and under supporting information file. 

Response to the reviewer’s comment

1. Comment 3: Have the authors made all data underlying the findings in their manuscript fully available?

Author Response: All the data are made available with in the manuscript except the one that are inside the supporting information file. 

2. Comment 4: Is the manuscript presented in an intelligible fashion and written in standard English?

Author Response: The manuscript has been revised, and corrected based on the standard format. 

3. Comment 5: Reviewer #1

- The manuscript is not presented in an intelligible fashion and is written in standard English. Authors need to read this paper carefully and make all the necessary changes, as there are several errors while reporting this paper.

Author Response: The manuscript has been revised and corrected making all the necessary changes.

4. Comment 5: Reviewer #2

- Abstract: A few clarifications needed. Please reword:

- Acute respiratory infection (ARI) is responsible for 30-50 percent of visits to health facility and 20-30 percent of admissions to hospital in Nepal. – is this for both adults and children or for children under 5 years old?

Author Response: The data is for under-five children. 

- What does it mean, ‘incidence of ARI/1000 children < 5 years of age is 344 in Nepal’? Does this mean 344/1000? Please reword: The incidence of ARI in children < 5 years old is 344/1000. The sentences are not clear. Please reword to make it clear.

Author Response: The sentence has been reworded to “Incidence of ARI among under-5 years of age is 344 per 1000 children in Nepal.”

- Non-probability is not equivalent to purposive. Purposive is to target those with ARI. In this study, I believe it is convenient sampling – please stick to convenient sampling.

Author Response: Changed to convenient sampling.

- When you say out of the 286 children, half had ARI. What did the other children have?

Author Response: It means, out of the total participants, half of them had ARI while others were visiting OPD for other cause. 

- Also, do not include conclusion that male children are not susceptible to ARI, because the results were insignificant.

Author Response: The sentence has been omitted. 

• Introduction

- Line 60: ARI is responsible for about 30-50% of visits…. Question: is this in children under 5 years old? Please specify.

Author Response: The data is for under-five children. The sentence has been reworded.

- Line 64: The deaths due to ARI at health facility is 127. What does this mean? Any health facility or hospitals or primary care? Can remove this information as it is not clear.

Author Response: The sentence has been removed.

- Methodology

- Line 91: Please specify the time and duration of the data collection period e.g. from month, year until month, year.

Author Response: The duration for data collection has been specified. 

- Line 106: Please add the Cronbach’s alpha for the pretesting that you have done, to show whether the questionnaire was reliable.

Author Response: The calculated value of Cronbach’s alpha was 0.85, which is mentioned in the “Instruments” section under methods.

- From paragraph 66-73 in the introduction, please add in the methodology the factors that was studied in your study.

Author Response: It has been mentioned as “variables under study” under methods. 

- Chi-squared test was done to show relationship between the categorical variables. My personal opinion is it is better to do simple logistic regression, and then move on to multiple logistic regression for the significant factors. Any reason why you use Chi-squared test rather than simple logistic?

Author Response: Simple logistic regression has been carried for bivariate analysis. 

- Discussion

- 218: In the discussion, you do not need to mention twice but to just discuss about the three factors that were associated with increase risk of ARI in children, which are religion, presence of child in the kitchen while cooking and presence of respiratory tract infection in family member.

Author Response: The factors has been discussed once.

---

## [Decision Letter · Decision Letter 1]

11 Mar 2022

Prevalence and Factors Associated with Acute Respiratory Infection among Under-five Children in Selected Tertiary Hospitals of Kathmandu Valley

PONE-D-21-19925R1

Dear Dr. Pratima Ghimire,

We’re pleased to inform you that your manuscript has been judged scientifically suitable for publication and will be formally accepted for publication once it meets all outstanding technical requirements.

Kind regards,

Sajid Bashir Soofi

Academic Editor

PLOS ONE

Additional Editor Comments (optional):

Congratulations, your paper is accepted for publication

Reviewers' comments:

Reviewer's Responses to Questions

**Comments to the Author**

1. If the authors have adequately addressed your comments raised in a previous round of review and you feel that this manuscript is now acceptable for publication, you may indicate that here to bypass the “Comments to the Author” section, enter your conflict of interest statement in the “Confidential to Editor” section, and submit your "Accept" recommendation.

Reviewer #2: All comments have been addressed

Reviewer #3: All comments have been addressed

2. Is the manuscript technically sound, and do the data support the conclusions?

Reviewer #2: Yes

Reviewer #3: Yes

3. Has the statistical analysis been performed appropriately and rigorously? 

Reviewer #2: Yes

Reviewer #3: Yes

4. Have the authors made all data underlying the findings in their manuscript fully available?

Reviewer #2: Yes

Reviewer #3: Yes

5. Is the manuscript presented in an intelligible fashion and written in standard English?

Reviewer #2: Yes

Reviewer #3: Yes

6. Review Comments to the Author

Reviewer #2: Thank you for addressing all of the comments. The manuscript could improve in its linguistic delivery. Otherwise, no other comments.

Reviewer #3: all the comments have been adequately addressed.

the paper is of significant public health importance and also opens the discussion on environmental health and impact .

7. PLOS authors have the option to publish the peer review history of their article (what does this mean?). If published, this will include your full peer review and any attached files.

Reviewer #2: No

Reviewer #3: **Yes: **Shabina Ariff Associate Professor ,Aga khan university Pakistan

---

## [Editor Report · Acceptance letter]

29 Mar 2022

PONE-D-21-19925R1 

Prevalence and factors associated with acute respiratory infection among under-five children in selected tertiary hospitals of Kathmandu Valley 

Dear Dr. Ghimire:

I'm pleased to inform you that your manuscript has been deemed suitable for publication in PLOS ONE. Congratulations! Your manuscript is now with our production department. 

Kind regards, 

on behalf of

Professor Sajid Bashir Soofi 

Academic Editor

PLOS ONE